# Association between Fat Distribution and Chronic Low Back Pain among 10,606 Adults: Data from the Korean National Health and Nutrition Examination Survey

**DOI:** 10.3390/ijerph19095599

**Published:** 2022-05-05

**Authors:** Minji Oh, Jongyeon Kim, Suji Lee, Seunghoon Lee, Jae-Dong Lee

**Affiliations:** 1Department of Clinical Korean Medicine, Graduate School, Kyung Hee University, Seoul 02447, Korea; begorgeous815@gmail.com (M.O.); dltnwlzzang@naver.com (S.L.); 2Department of Acupuncture and Moxibustion Medicine, Kyung Hee University Medical Center, Seoul 02447, Korea; jay.luke.kim@gmail.com (J.K.); ljdacu@gmail.com (J.-D.L.); 3Department of Acupuncture and Moxibustion, College of Korean Medicine, Kyung Hee University, Seoul 02447, Korea

**Keywords:** fat mass, fat distribution, obesity, chronic low back pain, KNHANES

## Abstract

Obesity is associated with chronic low back pain (CLBP), but the association between fat distribution and CLBP is unclear. This cross-sectional study evaluated the relationship using the Korean National Health and Nutrition Examination Survey data. A total of 10,606 adults (average age: 45.4, female: 57.1%) were included. We estimated the regional fat distribution, waist circumference, and body fat proportion, compared the values in people with and without CLBP, and stratified the estimates by sex and obesity status using a multivariable linear model. There were no statistically significant differences in the average waist circumference between the people with and without CLBP (*p* = 0.731) and the average fat proportion between those with and without CLBP (*p* = 0.731). The average regional fat distribution was significantly higher in the people with CLBP than in those without CLBP, in the upper limbs (11.4%, 95% confidence interval [CI]: [11.3, 11.5] vs. 11.2%, 95% CI: [11.1, 11.3], *p* < 0.05) and in the lower limbs (31.9%, 95% CI: [31.6, 32.2] vs. 31.4%, 95% CI: [31.2, 31.6], *p* < 0.01). More obvious among men, fat distribution in the lower limbs is higher than in people without obesity (*p* < 0.001). People with CLBP tend to have a higher fat distribution in the limbs than those without it and obese people with CLBP would need to reduce the fat in the lower limbs.

## 1. Introduction

Low back pain (LBP), a musculoskeletal disorder commonly observed in clinical practice, affects physical activity and deteriorates the quality of life [1]. One in ten people worldwide have LBP at any time point, and 70–85% of people have an LBP episode at some time in their life [2]. The costs associated with LBP are enormous, leading to a major economic burden for patients, the government, and health insurance companies [3]. As the etiology of LBP is complex, some factors related to its prevalence and incidence include the presence of comorbidities, lifestyle factors (e.g., smoking), occupational and psychosocial factors, physical inactivity [4,5], genetics [6], and sex [7]. Critical insights into the management and prevention of LBP may be gained with a more nuanced understanding of the related risk factors [3,8].

Obesity is associated with various musculoskeletal conditions and is accountable for serious disability and impaired quality of life [9]. Data on the association between obesity and LBP [10] are consistent across studies, in which the former was defined based on body mass index (BMI), which is most commonly used to determine adiposity. However, obesity status ascertained by BMI may not convey sufficient information on LBP, as BMI is a composite of various factors, such as fat mass and lean mass; additionally, it does not provide meaningful information on whole-body fat distribution. Thus, it may be relevant to investigate the relationship between fat distribution and LBP.

Some previous studies have focused on the association of fat mass distribution with LBP. Brady et al. provided evidence on the crucial role of fat mass, especially android fat relative to gynoid fat, in back disability and pain [11]. Although Hussain et al. showed that LBP-induced pain intensity and disability were related to body composition measures, such as fat mass, fat proportion, and waist circumference, they did not include sufficient information on how regional fat distribution is associated with LBP [12]. Another study found a weak association between high body fat mass accumulation levels around the hips and an increased prevalence of chronic low back pain (CLBP) in women [13]. None of the aforementioned studies quantified fat distribution using validated measurements.

Dual-energy X-ray absorptiometry (DXA) can overcome the limitations of prior studies. It is a valid and reliable technique for assessing body composition [14] and can validly describe the population-level distribution of body composition in people with obesity [15]. The Korean National Health and Nutrition Examination Survey (KNHANES) collected DXA measurement data on each body part (head, trunk, and upper/lower limbs) in 2008–2011. Several studies have analyzed the data of LBP and obesity using the KNHANES database. One study reported an association of LBP with abdominal obesity measured by waist circumstance among people aged 50 years or more using KNHANES V data, which does not include a questionnaire for CBLP and DXA data [16]. Another study examined whether BMI was a risk factor in CLBP patients with metabolic syndrome, not the association between CLBP and obesity [17]. However, to the best of our knowledge, no studies to date have examined the relationship between CLBP and regional fat distribution using DXA data. We assumed that waist circumference, body fat proportion against the total body weight, and fat distribution according to body regions, such as lower and upper limbs, would differ between patients with LBP and those without LBP. Accordingly, we attempted to examine the relationship between waist circumference, body fat proportion, regional fat distribution, and CLBP using the KNHANES IV data.

## 2. Materials and Methods

### 2.1. Data Sources and Study Population

The study population included participants of the KNHANES IV, 2008–2009, which comprised a health interview survey, a health examination survey, and a nutrition survey. The health interview survey included items pertaining to sociodemographic status and health-related behaviors and was used for obtaining data on morbidity, including self-reported CLBP status. The health examination survey comprised items on waist circumference and BMI as well as DXA measurements, from which we quantified regional fat mass.

Although the KHNAHES IV was initiated in 2007, the use of the DXA examination was started only in the second half of 2008; accordingly, we included observations from the period after 2008. The KNHANES V (2010–2012) also conducted DXA, but the health interview survey lacked any questions on prevalent CLBP. Therefore, our study population was limited to the individuals included in the KNHANES IV, 2008–2009.

In this population, we included people with data from both the health interview survey and health examination survey. We excluded those without DXA measurements or information on CLBP status. We also excluded people aged 18 years or younger, since the minimum age of those with CLBP data was 19 years, although the KNHANES IV targeted citizens aged 10 years or older.

### 2.2. Measurements

DXA uses an X-ray of two energy levels with different absorptivity to tissue components; in this manner, it quantifies fat mass, lean mass, and bone mass [18]. The KNHANES IV gathered data on the fat, bone, or lean mass of each body part (upper limbs, lower limbs, trunk, and head), as measured by DXA (Discovery-W fan-beam densitometer, Hologic Inc., Bedford, MA, USA). We combined the measurements of the left and right into one value and did not use the measurement of the head itself, except for the total mass.

With the DXA measurements, we examined two aspects of obesity: body fat proportion against total body weight and regional fat distribution against total body fat; the former for the measurement of relative fat mass and the latter for the assessment of how the total fat mass is distributed within the whole body.

The waist circumference represented the apparent fat distribution at the trunk and was derived from the health examination survey. Body fat proportion signifies the total fat mass relative to the total weight, while regional fat distribution pertains to the fat mass distribution in the upper and lower limbs, which we defined as the regional fat mass divided by the total fat mass. Data on body fat were obtained using DXA, and since the fat in the trunk measured by DXA included the pelvis and chest cavity as well as abdomen region, making it difficult to interpret clinically, regional fat distribution in the trunk was not included in the analysis.

### 2.3. Variables

Participants were assumed to have CLBP if they answered ‘yes’ to the following question in the health interview survey: “Have you experienced LBP persisting for three months or more in the most recent year?”

A multivariable linear model was employed to determine an association of CLBP with weight circumstance, and we adjusted several predefined covariates for sociodemographic factors (sex, age, education, occupation, household income, and region), health behaviors (smoking, drinking, and physical activity), and comorbidities (osteoporosis and depression). Region was included as a binary variable (Seoul vs. the other areas), although it originally had 16 levels in the data. Participants were considered smokers if they had a smoking habit at the time or had smoked five or more packs of cigarettes in their life. The variable for drinking identified people who had consumed one or more glasses of alcohol per month during the most recent year. For physical activity, we used three variables. One pertained to whether a participant partook in at least 20 min of vigorous physical activity three or more days a week, while another was related to participation in at least 30 min of moderate physical activity five or more days a week. The third variable was associated with walking for at least 30 min five or more days a week. Comorbidities were determined by the response to a question in the health interview survey about whether participants had experienced a condition for three or more months in the most recent year. Participants with a BMI higher or equal to 25 kg/m^2^ were assigned to the obesity group according to the guideline for obesity diagnosis in Korea [19].

To determine an association of CLBP with body fat proportion and regional fat distribution, we included the total lean body mass and the aforementioned covariates. This allowed us to obtain estimates conditional on the same total lean body mass. In other words, we tried to make estimates within people with similar body sizes except for fat mass.

### 2.4. Statistical Analysis

We employed linear models to estimate average waist circumferences, body fat proportions, or regional fat proportions by CLBP status. For each aim, we assumed three models. First, we only included the covariates as mentioned above for the estimation of averages by CLBP status conditional on the values of the covariates. The second model included an interaction term in the form of the BMI category and CLBP status to obtain estimates stratified by BMI category. For the third model, we added the interaction terms of BMI category, sex, and CLBP status for stratified estimates by sex and BMI category.

Following this, we calculated the marginal averages in each stratum. We first set sex, BMI category, or CLBP status to a value of each stratum and obtained predictions based on each model. Then, we averaged the predicted values, which is the marginal average of each stratum. For example, with the model including the interaction term of CLBP status and BMI category, we calculated the marginal average of either the waist circumference, body fat proportion, or regional fat distribution in every combination of CLBP status (yes vs. no) and BMI category (lower than 25 kg/m^2^ vs. 25 kg/m^2^ or higher).

We conducted a hypothesis test to compare the marginal averages in the CLBP and non-CLBP groups. Another hypothesis test was conducted to test for sex- or BMI category-related differences in the marginal averages between the two groups. We adjusted *p*-values from 40 hypothesis tests of our study to account for multiplicity, using the Benjamini–Hochberg method. This method controls the false discovery rate, the probability of positive results being false, under a certain threshold. Adjusted *p*-values less than 0.05 were considered significant.

To handle missing data, we employed the multiple imputation approach. Specifically, we imputed missing values using an R package, Amelia (version 1.7.6) [20] and created 10 imputed datasets. Using the imputed datasets, we performed the analyses, as described above, and combined the results, considering the variance among the estimates from the imputed datasets.

The K-NHANES IV provides weight variables so that the data represent the whole Korean population. To account for the survey design of the KNHANES IV, we used the ‘survey’ package (version 4.0) with R 4.0.2. [21,22].

## 3. Results

### 3.1. Characteristics of the Study Population

Of the 11,503 participants who completed the health interview and health examination survey, we excluded 268 participants due to the unavailability of data on DXA measurement or CLBP status. Then, we enrolled 10,606 participants after the exclusion of 629 patients aged < 19 years (Figure 1). The average age was 45.4 (standard deviation: 5.3) and 6059 participants were female (57.1%).

The weighted BMI average was 23.7 kg/m^2^ (95% confidence interval [CI]: [23.5, 23.9]) in the people with CLBP and 23.5 kg/m^2^ (95% CI: [23.4, 23.6]) in those without CLBP. The weighted average total lean mass was 42.3 kg (95% CI: [41.7, 42.9]) in the people with CLBP, which was lower than that in those without CLBP (46.5 kg, 95% CI: [46.3, 46.8]).

The people with CLBP tended to have a lower level of education and income than those without CLBP. The proportion of white-collar workers was lower in the CLBP group than in the non-CLBP group (10.5% vs. 24.7%). The non-CLBP group comprised participants who tended to be younger (43.2 years vs. 54.8 years), live in Seoul (24.4% vs. 12.7%) and be male compared to the CLBP group (47.5% vs. 67.6%).

There were fewer current smokers (18.2% vs. 28.1%) and drinkers (43.6% vs. 61.2%) in the CLBP group. In the non-CLBP group, a larger number of people participated in vigorous physical activity (18.6% vs. 17.1%), whereas only a small number of people showed moderate physical activity levels (13.3% vs. 18.2%) compared to the values in the CLBP group.

The people with CLBP tended to have depression (11.4% vs. 3.41%) and osteoporosis (13.3% vs. 2.10%) compared to those without it. Detailed estimates are summarized in Table 1.

### 3.2. Aim 1: Waist Circumference by CLBP Status

Overall, there were no statistically significant differences in the estimated marginal average waist circumference values between the people with and without CLBP. The marginal average estimated in the people with obesity was 90.1 cm (95% CI: [89.5, 90.7]) in the CLBP group and 89.9 cm (95% CI: [89.5, 90.3]) in the non-CLBP group; the difference was not statistically significant. The marginal average estimated in the people without obesity was 76.6 cm (95% CI: [76.0, 77.2]) in the CLBP group and 76.5 cm (95% CI: [76.2, 76.8]) in the non-CLBP group; the difference was not statistically significant. (Table 2).

### 3.3. Aim 2: Body Fat Proportion by CLBP Status

The estimated marginal average of fat proportion against the total body weight did not significantly differ by CLBP status. When this value was estimated within people with obesity, it was found to be statistically significantly lower in the people with CLBP than in those without it (30.7% vs. 31.3%, *p* < 0.05). This pattern was more obvious in men than in women. In men with obesity, the estimated marginal average of body fat proportion was lower in the CLBP group (25.7%, 95% CI: [24.8, 26.5]) than in the non-CLBP group (26.6%, 95% CI: [26.0, 27.3]); the difference was statistically significant (*p* < 0.05). On the other hand, in women with obesity, there was no statistically significant difference in the comparison of the CLBP and non-CLBP groups (*p* = 0.133). However, we could not identify statistically significant differences among the people without obesity (Table 3).

### 3.4. Aim 3: Regional Fat Distribution vs. CLBP Status

The estimated marginal average of the regional fat distribution in the lower limbs was significantly higher in the people with CLBP (31.9%, 95% CI: [31.6, 32.2]) than in those without CLBP (31.4%, 95% CI: [31.2, 31.6], *p* < 0.01). This pattern was more obvious in people with obesity. Among those with obesity, the estimated average was 30.2% (95% CI: [29.8, 30.6]) in the CLBP group and 29.4% (95% CI: [29.1, 29.6]) in the non-CLBP group (*p* < 0.01). The same pattern was observed in both sexes (Table 4).

The estimate in the upper limbs was statistically significantly higher in the people with CLBP (11.4%, 95% CI: [11.3, 11.5]) than in those without CLBP (11.2%, 95% CI: [11.1, 11.3], *p* < 0.05). Among men without obesity, the people with CLBP had the higher estimated average regional fat proportion in the upper limbs (10.5%, 95% CI: [10.3, 10.7]) than those without CLBP (10.2%, 95% CI: [10.0, 10.3], *p* < 0.05), but not in the women without obesity or in people with obesity (Table 4).

## 4. Discussion

The findings of our study, which included a nationally representative adult population from South Korea, suggest that the significance of regional fat mass distribution is high, compared to that of waist circumference and body fat proportion, in the context of CLBP. People with CLBP tend to have a higher fat distribution in the limbs than those without it. Focusing on people with obesity, the regional fat distribution in the lower limbs is higher than those without obesity. This tendency was more obvious among men than among women.

Some studies have demonstrated the presence of a relationship between waist circumference and CLBP. One study suggested that the likelihood of LBP development increases significantly in women with a high waist circumference [23], while another showed that LBP was related to a higher waist–hip ratio and BMI in men, with a stronger trend toward having a higher fat mass index [24]. However, our study that used KNHANES IV data could not identify a significant relationship between waist circumference and CLBP, except in men with obesity.

Some previous studies focused on the distribution of fat mass to investigate the relationship with LBP. A population-based longitudinal study showed that fat mass and fat distribution were affirmatively correlated with the intensity and disability associated with back pain [12]. Another found that higher levels of fat mass deposition around the hips were weakly related to increases in the prevalence of CLBP in women through the investigation of 1128 female twins [13]. Evidence exists on the important role of fat mass, specifically android fat relative to gynoid fat, in back pain and disability [11]. We found that the total fat mass relative to the total body weight did not differ by CLBP status, except in men with obesity; however, people with CLBP had a higher fat distribution in the upper and lower limbs than those without CLBP. Specifically, among people with obesity, the regional fat distribution in the lower limbs is higher in people with CLBP; otherwise, among people without obesity, the regional fat distribution in the upper limbs was higher in the CLBP group. These patterns were more distinct in men than in women. These results imply that the regional fat distribution in the limbs may convey more important information on CLBP than waist circumference and fat proportion in people regardless of sex.

Few studies have examined how fat distribution affects CLBP. One study proposed that regional fat distribution may play an important role in the etiology of adiposity-related disease [25]. Another study demonstrated the relationship between fat mass and LBP by suggesting that some systemic and biomechanical factors related to obesity contribute to the pathogenesis of LBP [12]. In this light, certain confounding factors associated with regional fat distribution in the limbs may play a major role in the pathogenesis of CLBP. Increased intermuscular adipose tissue (IMAT) in the lower limbs may be a possible factor in the association of CLBP with regional fat distribution among people with obesity. IMAT is detected under the muscle fascia and is usually considered ectopic fat. Muscle injury, obesity, inactivity, disease status, and age are factors related to increased IMAT values. These increases in the IMAT levels may also cause a larger number of muscle, metabolic, and mobility dysfunctions [26]. For example, a previous study examined the relationship between IMAT and lower limb muscle strength and power in older adults [27]. If the fat distribution in the lower limbs increases in association with obesity, the probability of IMAT occurrence in the lower limb muscles too may rise, which may lead to lower extremity muscle weakness and susceptibility to CLBP. Some studies have examined the relationship between muscle power in the lower extremities and LBP. One study emphasized that muscle weakness and an imbalance in the muscle power ratio of the quadriceps femoris and hamstring play important roles in the prevalence of LBP along with muscle weakness and imbalance in the lumbar muscles [28]. Another showed that the hamstring tends to be particularly weak in patients with LBP [29].

A Spanish twin study suggested that early-life environmental or genetic factors may confound the association between the waist–hip ratio and CLBP [13]. Considering the similar patterns observed in this study, our results may have arisen from confounders related to both fat distribution in the lower limbs and the pathogenesis of CLBP. Of note, while the pattern we observed was obvious in the male participants with obesity, the aforementioned Spanish twin study targeted only women. The pattern may have been more visible in men due to differences in the body composition between men and women. Sex differences in the fat phenotypes are presumably established by a complicated interaction of genetic, epigenetic, and hormonal factors [30]. Women tend to accumulate a large amount of adipose tissue in the hips and thighs and have a higher percentage of body fat values compared to men. These basic differences in physical characteristics between the sexes are thought to influence the variations in the distribution of fat in the lower limbs between men and women. Further studies should focus on how sex and race affect the association between fat distribution and CLBP pathogenesis.

This study had some limitations. First, while DXA is a useful tool for determining body composition, it cannot accurately distinguish between different types of fat, such as subcutaneous fat and visceral fat. For this reason, it was challenging to discuss further which type of fat tissue was associated with the patterns observed in this study. Second, for people without obesity, especially in men, we observed more fat in the upper limbs but could not suggest an appropriate physiological interpretation due to the lack of previous research on the relation between the fat in the upper limbs and CLBP. Third, we could not include information on the physiological state related to body fat distribution, such as menopause, to explain the difference between male and female sex. Fourth, clinically important differences in fat distribution between patients with CLBP and non-CLBP were unknown, and the regional fat difference of the lower limbs in our study did not seem to be large enough. Fifth, since DEX could not accurately measure the fat in the abdominal region, which is likely to be related to CLBP [31], the correlation between abdominal adiposity and CLBP could not be investigated in this study. Sixth, CLBP was not measured as a validated measurement, but simply as a questionnaire about pain experience through a health interview survey in the K-NHANES IV.

Nevertheless, this study has the following strengths. First, it investigated the relationship between fat distribution and CLBP using the large population data, so the results represent the general population of Korea. In addition, rather than simply examine the relationship between the absolute amount of fat mass and CLBP, we explored the regional fat distribution. To the best of our knowledge, this study is the first to demonstrate the relationship between the fat distribution in the upper and lower limbs and CLBP.

Clinically, in the case of CLBP patients, it may be helpful to provide treatment and encourage participation in exercise to reduce the fat percentage, especially in patients with a BMI of 25 kg/m^2^ or higher in the lower body, rather than simply focusing on weight loss. However, since it is very difficult to reduce only the fat in the lower limbs, patients can aim to reduce overall body fat through whole-body exercise and simultaneously strengthen the lower body muscles through lower body focused workouts for decreasing relative fat distribution in the lower limbs.

## 5. Conclusions

In this nationwide representative sample of Korean adults, we found that people with CLBP tend to have a higher fat distribution in the limbs than those without CLBP; this pattern was particularly evident in our male participants. Our findings suggest that the fat distribution in the lower limbs provides a greater degree of information on CLBP than simple waist circumference in people with obesity. Future large-scale cohort studies or clinical trials should confirm whether fat mass reductions in the lower limbs can prove effective in CLBP treatment.

## Figures and Tables

**Figure 1 ijerph-19-05599-f001:**
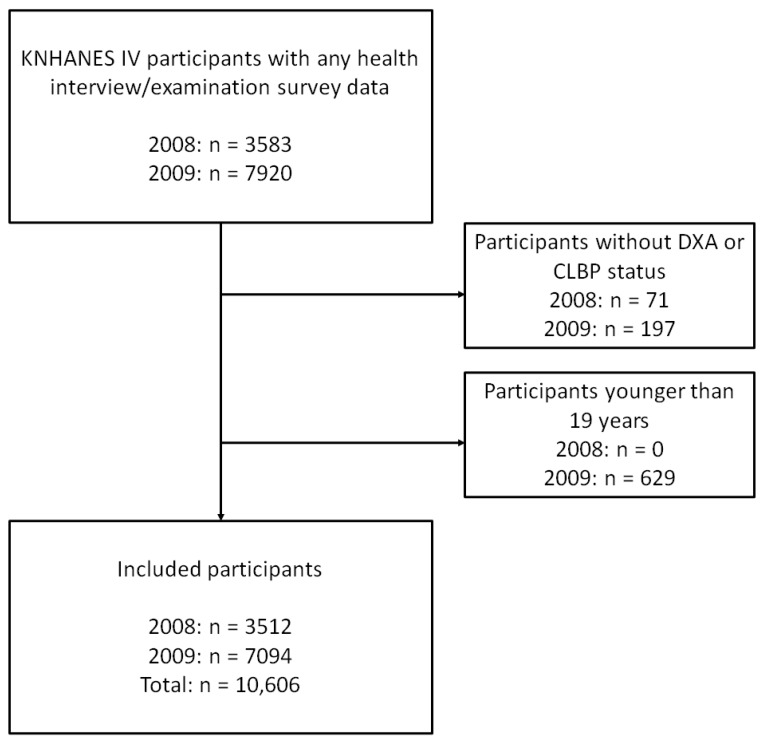
Flow chart of participant enrollment. Abbreviations: KNHANES, Korean National Health and Nutrition Examination Survey; DXA, dual-energy X-ray absorptiometry; CLBP, chronic low back pain.

**Table 1 ijerph-19-05599-t001:** Characteristics of the Study Population According to CLBP Status (KNHANES IV, 2008–2009).

	CLBP (n = 1981)		No CLBP (n = 8625)	
	Unweighted Values ^a^	Weighted Estimates [95% CI] ^b^	Unweighted Values ^a^	Weighted Estimates [95% CI] ^b^
Anthropometric factors				
BMI (kg/m^2^)	23.7 (21.5, 25.8)	23.7 [23.5, 23.9]	23.3 (21.2, 25.6)	23.5 [23.4, 23.6]
Total lean mass (kg)	39.2 (35.3, 46.0)	42.3 [41.7, 42.9]	43.3 (37.3, 52.7)	46.5 [46.3, 46.8]
Sociodemographic factors				
Age (year)	63 (48, 71)	54.8 [53.6, 55.9]	45 (34, 59)	43.2 [42.6, 43.8]
Sex				
Female	1422 (71.8)	67.6 [65.3, 69.9]	4637 (53.8)	47.5 [46.6, 48.5]
Education				
Elementary	1113 (56.2)	45.2 [42.0, 48.4]	1897 (22.0)	16.0 [14.7, 17.2]
Middle	231 (11.7)	12.2 [10.5, 14.0]	961 (11.1)	10.1 [9.30, 10.8]
High	423 (21.4)	28.3 [25.2, 31.5]	3255 (37.7)	41.8 [40.1, 43.4]
College	214 (10.8)	14.2 [11.8, 16.6]	2478 (28.7)	32.2 [30.3, 34.1]
NA	0 (0)	-	34 (0.394)	-
Occupation				
White-collar	142 (7.17)	10.5 [8.63, 12.3]	1804 (20.9)	24.7 [23.3, 26.1]
Blue-collar	867 (43.8)	41.0 [37.9, 44.2]	3465 (40.2)	39.1 [37.4, 40.9]
Unemployed	967 (48.8)	48.5 [45.4, 51.6]	3294 (38.2)	36.2 [34.8, 37.5]
NA	5 (0.252)	-	62 (0.719)	-
Income				
Low	731 (36.9)	29.5 [26.7, 32.3]	1473 (17.1)	13.9 [12.6, 15.2]
Low–mid	461 (23.3)	23.4 [20.8, 25.9]	2098 (24.3)	24.4 [22.7, 26.0]
Mid–high	427 (21.6)	25.0 [22.1, 27.8]	2469 (28.6)	30.7 [29.0, 32.5]
High	346 (17.5)	22.2 [19.4, 25.0]	2448 (28.4)	31.0 [28.6, 33.4]
NA	16 (0.808)	-	137 (1.59)	-
Region				
Seoul	151 (7.62)	12.7 [10.1, 15.4]	1600 (18.6)	24.4 [22.1, 26.8]
Health Behavior				
Currently smoking				
Yes	286 (14.4)	18.2 [16.0, 20.4]	2066 (24.0)	28.1 [27.0, 29.3]
NA	6 (0.303)	-	35 (0.406)	-
Drinking				
Yes	748 (37.8)	43.6 [40.8, 46.4]	4859 (56.3)	61.2 [59.9, 62.5]
NA	3 (0.151)	-	29 (0.336)	-
Physical activity				
Vigorous	326 (16.5)	17.1 [15.1, 19.2]	1479 (17.1)	18.6 [17.4, 19.7]
NA	3 (0.151)	-	34 (0.394)	-
Moderate	379 (19.1)	18.2 [15.9, 20.5]	1204 (14.0)	13.3 [12.3, 14.4]
NA	4 (0.202)	-	38 (0.441)	-
Walking	916 (46.2)	47.3 [44.3, 50.3]	4004 (46.4)	46.7 [45.4, 48.1]
NA	4 (0.202)	-	40 (0.464)	-
Comorbidity				
Depression	234 (11.8)	11.4 [9.61, 13.1]	331 (3.84)	3.41 [2.93, 3.88]
Osteoporosis	336 (17.0)	13.3 [11.6, 15.1]	264 (3.06)	2.10 [1.80, 2.40]

Abbreviations: KNHANES, Korean National Health and Nutrition Examination Survey; CLBP, chronic low back pain; CI, confidence interval; BMI, body mass index; NA, not available. ^a^ Each unweighted value is either a median with an interquartile range or a count with its percentage. ^b^ Each weighted estimate is a counterpart of each unweighted one and was estimated from 10 imputed datasets. For nominal variables, it is an estimated percentage; for continuous variables, it is an estimated average, with 95% confidence intervals.

**Table 2 ijerph-19-05599-t002:** Estimated Marginal Average of Waist Circumference According to CLBP Status.

BMI, kg/m^2^	Sex	CLBP (cm, [95% CI])	No CLBP (cm, [95% CI])	*p*-Value ^a^	Adjusted *p*-Value ^a^	P for Interaction ^b^	Adjusted P for Interaction ^b^
≥25	Female	88.3 [87.5, 89.2]	87.9 [87.3, 88.5]	0.328	0.505	-	
	Male	91.1 [90.1, 92.1]	92.2 [91.7, 92.7]	0.0493	0.123	0.0440	0.123
	Overall	90.1 [89.5, 90.7]	89.9 [89.5, 90.3]	0.640	0.731	-	
<25	Female	74.3 [73.6, 75.0]	73.8 [73.4, 74.2]	0.127	0.254	-	
	Male	78.9 [77.9, 79.9]	79.2 [78.9, 79.6]	0.470	0.627	0.139	0.265
	Overall	76.6 [76.0, 77.2]	76.5 [76.2, 76.8]	0.702	0.762	0.908	0.931
Overall		80.8 [80.3, 81.3]	80.7 [80.3, 81.0]	0.606	0.731	-	

Abbreviations: CLBP, chronic low back pain; CI, confidence interval; BMI, body mass index. ^a^ The *p*-values are used to test the hypothesis that the marginal averages of the two groups are the same. ^b^ The *p*-values are considered for testing the hypothesis that the differences of the marginal averages estimated in the CLBP and non-CLBP groups are the same, on comparing the female and male participants, or a BMI lower than 25 kg/m^2^ and that higher than or equal to 25 kg/m^2^.

**Table 3 ijerph-19-05599-t003:** Estimated Marginal Average of Fat Proportion against Body Weight by CLBP Status.

BMI, kg/m^2^	Sex	CLBP (%, [95% CI])	No CLBP (%, [95% CI])	*p*-Value ^a^	Adjusted *p*-Value ^a^	P for Interaction ^b^	Adjusted P for Interaction ^b^
≥25	Female	35.4 [34.9, 35.9]	35.9 [35.5, 36.3]	0.0598	0.133	-	
	Male	25.7 [24.8, 26.5]	26.6 [26.0, 27.3]	0.00784	0.0333	0.256	0.445
	Overall	30.7 [30.2, 31.2]	31.3 [30.9, 31.7]	0.0107	0.0389	-	
<25	Female	28.9 [28.4, 29.4]	28.8 [28.4, 29.3]	0.705	0.762	-	
	Male	20.3 [19.6, 21.0]	20.0 [19.5, 20.4]	0.279	0.465	0.513	0.662
	Overall	24.6 [24.2, 25.0]	24.4 [24.2, 24.7]	0.448	0.618	0.00833	0.0333
Overall		26.5 [26.1, 26.9]	26.6 [26.3, 26.8]	0.631	0.731	-	

Abbreviations: CLBP, chronic low back pain; CI, confidence interval; BMI, body mass index. ^a^ The *p*-values considered are to test the hypothesis that the marginal averages of the two groups are the same. ^b^ The *p*-values are considered for testing the hypothesis that the differences of the marginal averages estimated in the CLBP and non-CLBP groups are the same, on comparing female and male participants, or BMI lower than 25 kg/m^2^ and that higher than or equal to 25 kg/m^2^.

**Table 4 ijerph-19-05599-t004:** Estimated Marginal Average of Fat Distribution against Total Body Fat in Each Body Part.

Body Part	BMI, kg/m^2^	Sex	CLBP (%, 95% CI)	No CLBP (%, 95% CI)	*p*-Value ^a^	Adjusted *p*-Value ^a^	P for Interaction ^b^	Adjusted P for Interaction ^b^
Upper limbs	≥25	Female	12.5 [12.3, 12.7]	12.4 [12.3, 12.6]	0.373	0.553	-	
		Male	10.3 [9.96, 10.6]	10.1 [9.99, 10.3]	0.401	0.573	0.818	0.861
		Overall	11.4 [11.2, 11.6]	11.3 [11.1, 11.4]	0.104	0.219	-	
	<25	Female	12.3 [12.1, 12.5]	12.2 [12.1, 12.4]	0.324	0.505	-	
		Male	10.5 [10.3, 10.7]	10.2 [10.0, 10.3]	0.00242	0.0138	0.0493	0.123
		Overall	11.4 [11.2, 11.5]	11.2 [11.1, 11.3]	0.0217	0.0723	0.947	0.947
Overall			11.4 [11.3, 11.5]	11.2 [11.1, 11.3]	0.00712	0.0333	-	
Lower limbs	≥25	Female	31.4 [30.9, 31.8]	30.2 [29.8, 30.6]	3.12 × 10^−6^	0.000125	-	
		Male	29.6 [28.9, 30.3]	28.2 [27.9, 28.6]	7.88 × 10^−5^	0.00105	0.592	0.731
		Overall	30.2 [29.8, 30.6]	29.4 [29.1, 29.6]	0.000110	0.00110	-	
	<25	Female	34.1 [33.6, 34.6]	34.4 [34.1, 34.8]	0.197	0.358	-	
		Male	31.2 [30.7, 31.7]	30.2 [29.9, 30.4]	4.93 × 10^−5^	0.000986	0.000187	0.00150
		Overall	32.6 [32.3, 33.0]	32.3 [32.1, 32.5]	0.0600	0.133	0.0430	0.123
Overall			31.9 [31.6, 32.2]	31.4 [31.2, 31.6]	0.000891	0.00594	-	

^a^ The *p*-values are to test the hypothesis that the marginal averages of the two groups are the same. ^b^ The *p*-values are to test the hypothesis that the differences of the marginal averages estimated in the CLBP and non-CLBP groups are the same, on comparing female and male participants, or BMI lower than 25 kg/m^2^ and that higher than or equal to 25 kg/m^2^.

## Data Availability

The datasets of this study are publicly available, and the analyzed data are available from the corresponding author on reasonable request: https://knhanes.kdca.go.kr/knhanes/sub03/sub03_02_05.do (accessed on 1 December 2019).

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
