# Peer review of "Association between Fat Distribution and Chronic Low Back Pain among 10,606 Adults: Data from the Korean National Health and Nutrition Examination Survey"

_ijerph, 2022, doi:10.3390/ijerph19095599_

Round 1

Reviewer 1 Report

The authors used a very rich database on a large population to describe the relationship between chronic low back pain (CLBP) and fat distribution, while trying to address the limitations of previous studies on measuring fat distribution. One limitation highlighted by the authors in previous studies is the use of unvalidated methods to measure fat distribution and dissociate it from BMI. This study took this limitation into account, but still uses an unvalidated method to measure CLBP. There is the Chronic Pain Grade Questionnaire (CPGQ), which is a reliable and valid instrument for use in population surveys of LBP. The authors should have at least discussed this limitation. That said, I think the authors conducted a good study, but there are still many inaccuracies that make replication difficult.

In the abstract, please add the information on the statistical tests that led to the results.

Lines 64 & 65: Reference 17 included only participants with CLBP, it cannot conclude an association between CLBP and metabolic syndrome. For that, it would have been necessary to compare the distribution of SM between CLBP and non CLBP using a statistical test. As for reference 16, if it has already explored the association of CLBP with waist circumference, why repeat that analysis here in the same population? More precision are needed for this part of the introduction.

The section on DXA measurements and the section on outcomes (line 102) should be merged. DXA is only the method used to measure obesity parameters. Then the aims should be at the end of the introduction and not in the outcome section. This would allow for understanding of each of the outcomes detailed in the method section.

The section on exposure and covariates is confusing. There should be a description of the covariates and how they were measured. The specification of the multivariate model should be in the statistical section. Before discussing the variable included in each model, it should already be stated what type of statistical analysis was performed. 

A linear regression model was used for all analyses, did the authors explore a potential deviation from linearity? a non-linear relationship?

Line 161: It should be made clear what stratification weights/variables were considered to take into account the study design.

It is important to be able to adjust the analyses on several potential confounding factors, it is also important to report how these variables were chosen (a priori and or a posteriori?)

Author Response

Point 1: The authors used a very rich database on a large population to describe the relationship between chronic low back pain (CLBP) and fat distribution, while trying to address the limitations of previous studies on measuring fat distribution. One limitation highlighted by the authors in previous studies is the use of unvalidated methods to measure fat distribution and dissociate it from BMI. This study took this limitation into account, but still uses an unvalidated method to measure CLBP. There is the Chronic Pain Grade Questionnaire (CPGQ), which is a reliable and valid instrument for use in population surveys of LBP. The authors should have at least discussed this limitation. That said, I think the authors conducted a good study, but there are still many inaccuracies that make replication difficult.

Response 1: We also agree with the reviewer's point. However, the K-NHANES IV, which is the database used in our study, was not planned to measure CLBP through a validated questionnaire like CPGQ. This limitation was described in the discussion (line 331), and in future studies, we will use a validated questionnaire for measuring CLBP.

Point 2: In the abstract, please add the information on the statistical tests that led to the results.

Response 2: As reviewer's comments, we added the information on the statistical details in the abstract.

Point 3: Lines 64 & 65: Reference 17 included only participants with CLBP, it cannot conclude an association between CLBP and metabolic syndrome. For that, it would have been necessary to compare the distribution of SM between CLBP and non CLBP using a statistical test. As for reference 16, if it has already explored the association of CLBP with waist circumference, why repeat that analysis here in the same population? More precision are needed for this part of the introduction.

Response 3: References 16 and 17 are studies that analyzed the data on LBP and obesity using the KNHANES database. As mentioned by the reviewer, inaccurate information was described in the manuscript, so we correct these as follows. Reference 16 did not analyze DXA data, and it used KNHANES V (2010-2012) data, which examed only "LBP," not "chronic LBP." Therefore, in our study, KNHANES IV (2008-2009) data, which included data for "CLBP" and DXA, was used. We change the CLBP to LBP in the manuscript. Also, in reference 17, the association between CLBP and metabolic syndrome was not determined, so we deleted this in the manuscript (line 62).

Point 4: The section on DXA measurements and the section on outcomes (line 102) should be merged. DXA is only the method used to measure obesity parameters. 

Response 4: As reviewer's comments, we merged the section on DXA measurements and outcomes to the Measurements.

Point 5: Then the aims should be at the end of the introduction and not in the outcome section. This would allow for understanding of each of the outcomes detailed in the method section.

Response 5: The word "AIM" we described seems to be confusing. Our intention for describing these was how to define and measure 'waist circumstance,' 'body fat proportion,' and 'regional fat distribution'. Therefore, the terms of 'aims' were deleted, and only the description for each measurement was described in the Measurements.

Point 6: The section on exposure and covariates is confusing. There should be a description of the covariates and how they were measured. 

Response 6: In this study, we did not examine the causal relationship between CLBP and obesity. Instead of, we analyzed the differences in obesity(waist circumstance, body fat proportion, regional fat distribution) according to LBP, BMI category, and gender using cross-sectional data (K-NHANES IV). Therefore, the term 'exposure' was deleted and changed to the term 'variable' (line 112). As for covariates, factors expected to affect obesity were defined in advance among the measurements in the K-NHANES IV database (2.3. Variables).

Point 7: The specification of the multivariate model should be in the statistical section. Before discussing the variable included in each model, it should already be stated what type of statistical analysis was performed. 

Response 7: We first described what variables we took from the survey data and how they were measured. Then, we described the specifications of the models in the 'statistical analysis' part. However, as reverer's suggestion, we additionally described the model's specification, 'multivariable linear model,' before discussing the variables in the Variables (line 116).

Point 8: A linear regression model was used for all analyses, did the authors explore a potential deviation from linearity? a non-linear relationship?

Response 8: We appreciate that the reviewer questioned the linearity assumption of our models. We did consider options like splines and categorization of continuous variables, but we assumed linearity because most of the variables were categorical. Thus, we thought the potential deviation might not be so huge as to hurt the validity of our models.

Point 9: Line 161: It should be made clear what stratification weights/variables were considered to take into account the study design.

Response 9: The K-NHANES IV provides weight variables to make the data representative of the whole Korean population. We included the variables in our modeling process using the ‘survey’ package of R and described them in the manuscript (line 164).

Point 10: It is important to be able to adjust the analyses on several potential confounding factors, it is also important to report how these variables were chosen (a priori and or a posteriori?)

Response 10: Predefined covariates expected to affect the results were adjusted in Model 1, and BMI category and gender were stratified in Model 2 and 3. We described this clearly in the manuscript (line 117).

Reviewer 2 Report

This study compares estimates of waist circumference, percentage body fat and body fat distribution between people with and without chronic low-back pain (CLBP) using survey and dual-energy X-ray absorptiometry (DXA) data. Given the reported association between obesity and increased BMI and CLBP, it is highly relevant to further investigating the details of such association. The manuscript is clearly written. The methods and results are well explained and reported. The discussion is well focused on the study findings and limitations are identified. Thus, I recommend the paper for publication and I leave the following suggestions.

  1. Indication of the sample age range and sex distribution could be given in the Abstract and Methods sections.
  2. Information about correction for multiple pairwise comparisons should be given when presenting the statistical methods.
  3. The meaning of “effective measure modification (EMM)” (tables 2, 3 & 4) should be made clear.

Author Response

Point 1: Indication of the sample age range and sex distribution could be given in the Abstract and Methods sections.

Response 1: As reviewer's comments, we added the total sample age range and sex distribution in the Abstract and Results section (3.1. Characteristics of the study population).

Point 2: Information about correction for multiple pairwise comparisons should be given when presenting the statistical methods.

Response 2: We did not account for multiplicity in the manuscript. We added adjusted p-values using the Benjamini-Hochberg method, and we considered adjusted p-values less than 0.05 significant. After adjusting, each result of the analysis was not changed. We added lines describing the method, and changed some statistics in the manuscript (Abstract; 2.5. Statistical analysis; Results; Table 2,3,4). 

Point 3: The meaning of “effective measure modification (EMM)” (tables 2, 3 & 4) should be made clear.

Response 3: We used the term, effect measure modification (EMM), for a situation that the measures of effect (or association, in our case) are modified by levels of a variable. Traditionally, interaction is a widely used term in the medical field, so we modified our manuscript accordingly (Table 2,3,4).

Round 2

Reviewer 1 Report

Thank you for taking into account some of the points highlighted. I think that the way the statistical method part, as well as the use of weights to have a representative sample, are not clear and make it difficult to reproduce these analyses. I have no further questions and/or suggestions, except that it is useful to report after each answer, the changes made in the manuscript. It is more practical and it would avoid indicating line numbers that do not correspond to the changes made, which is the case here.